

# JacLy: a Jacobian-based method for the inference of metabolic interactions from the covariance of steady-state metabolome data

Mohammad Jafar Khatibipour[1,2], Furkan Kurtoğlu[1] and Tunahan Çakır[1]

[1] Computational Systems Biology Group, Department of Bioengineering, Gebze Technical University, Gebze, Kocaeli, Turkey

[2] Department of Chemical Engineering, Gebze Technical University, Gebze, Kocaeli, Turkey

## ABSTRACT

Reverse engineering metabolome data to infer metabolic interactions is a challenging research topic. Here we introduce JacLy, a Jacobian-based method to infer metabolic interactions of small networks (<20 metabolites) from the covariance of steady-state metabolome data. The approach was applied to two different *in silico* small-scale metabolome datasets. The power of JacLy lies on the use of steady-state metabolome data to predict the Jacobian matrix of the system, which is a source of information on structure and dynamic characteristics of the system. Besides its advantage of inferring directed interactions, its superiority over correlation-based network inference was especially clear in terms of the required number of replicates and the effect of the use of priori knowledge in the inference. Additionally, we showed the use of standard deviation of the replicate data as a suitable approximation for the magnitudes of metabolite fluctuations inherent in the system.

## INTRODUCTION

Inference of cellular interactions by processing biomolecular data is a widely-used approach to investigate functional properties of cellular systems. Perturbations due to genetic/environmental alterations and diseases lead to changes in functionality due to change in cellular network structure, and network inference using the biomolecular data of the perturbation states uncovers the changes in network structure. When applied to the data of metabolite levels, the approach infers metabolic interaction (*Srividhya et al., 2007*; *Çakır et al., 2009*; *Hendriks et al., 2011*; *Çakır & Khatibipour, 2014*). The general trend is to use dynamic data to infer directed metabolic networks, and steady-state data to infer undirected networks. On the other hand, there are approaches that infer directed metabolic interactions from steady-state metabolome data by utilizing inherent intrinsic variability in such data (*Steuer et al., 2003*; *Öksüz, Sadıkoğlu & Çakır, 2013*).

Corresponding author
Tunahan Çakır, tcakir@gtu.edu.tr, tcakir@gmail.com

While the principles of conservation of mass and conservation of energy set the boundaries for deterministic behavior of the metabolic network, the inherent randomness in this network, as it exists in other biological networks, leads to small variability in the steady-states of the system at equivalent macroscopic conditions (*Wu et al., 2005*; *Kresnowati et al., 2006*). From a microscopic point of view, the inherent randomness is believed to be the result of existence of discrete particles in the system, and molecular fluctuations are inherent in the mechanism by which the system evolves (*Kampen, 1992*). Continuous change in micro-environment as well as multilevel complex regulatory mechanisms in the metabolic network are also the causes of observed variability in the steady-states of the system (*Steuer et al., 2003*). Although this intrinsic randomness introduces a great obstacle and difficulty in modeling and simulation of metabolic networks, at the same time it provides an opportunity to infer and estimate the active metabolic network at a specific condition/context just by reverse engineering the corresponding replicates of metabolome data. Considering the fact that information on interactions between metabolites and hence the structure of the active metabolic network is implicit in these data, the main questions are (i) how much information on the structure of the network is hidden in the data, and (ii) how we can extract as much as possible of that information.

One common approach that utilizes inherent variability in steady-state data is correlation based inference methods, especially the Gaussian Graphical Model (GGM) approach (*Çakır et al., 2009*; *Montastier et al., 2015*; *Wang et al., 2016*). Correlation based approaches are capable of detecting strong interactions in the metabolic network to some extent. However, they infer interactions only in undirected manner, and they have limited power in the detection of weak interactions (*Çakır et al., 2009*). A directed network inference approach from steady-state metabolome data is also available in the literature (*Steuer et al., 2003*; *Öksüz, Sadıkoğlu & Çakır, 2013*; *Çakır & Khatibipour, 2014*). The approach is based on the prediction of interaction strengths from the covariance of the data. The network structure information stored in the inherent variability of the data is reflected on the covariance of the data, and later used in the prediction of interaction strengths in the form of a Jacobian matrix. The Jacobian matrix of a cellular interaction system contains a significantly high amount of valuable information both on the structure and dynamic characteristics of the system. Numbers in this matrix easily provide us with detailed information on the underlying interactions in the network, such as direction of interactions, nature of interactions (positive or negative effects), and strengths of interactions (*Steuer et al., 2003*; *Öksüz, Sadıkoğlu & Çakır, 2013*). The Lyapunov equation provides a link between the Jacobian matrix of a cellular system and the covariance matrix of the replicates of steady-state data. This equation is the result of a Langevin type approach for the description of stochastic processes at macroscopic level (*Scott, 2013*). The Lyapunov equation was also used previously to infer differential changes in a Jacobian matrix rather than the inference of the network structure itself (*Sun & Weckwerth, 2012*; *Kügler & Yang, 2014*). In another work (*Sun, Länger & Weckwerth, 2015*), a comparison of several least square and regularization methods in solving the Lyapunov equation for the Jacobian matrix is provided. However, in that work, the structure of the Jacobian (zero and non-zero

elements) is specified *a priori* by the stoichiometric matrix of the metabolic network. Therefore, the problem is reduced to the estimation of magnitudes for non-zero elements in the Jacobian matrix, which might not be considered as a network inference.

In our previous work, we focused on a theoretical analysis on the applicability of Jacobian-based inference of directed metabolic interactions from steady-state data (*Öksüz, Sadıkoğlu & Çakır, 2013*). *In silico* covariance data were used to predict a Jacobian matrix with an optimization framework based on dual objective function. The objective function was simultaneous maximization of the sparse structure, as observed in cellular networks (*Yeung, Tegnér & Collins, 2002*; *Tegnér & Björkegren, 2007*; *Bordbar et al., 2014*; *Shao et al., 2015*; *Mangan et al., 2016*), and minimization of the residual norm of the Lyapunov equation (see 'methods'). The theoretical analyses proved the potential usefulness of the approach. Here, we improve the algorithm used in that work considerably in terms of speed and optimality, and apply it successfully to *in silico* metabolome data.

## METHODS

### Problem definition

Provided that replicates of metabolome data are available for an organism in a specific condition, and considering the fact that information on interactions between metabolites and hence the structure of the metabolic network is implicit in these data, the problem is to extract from the data as much knowledge as possible to infer the active metabolic network in that condition. A metabolic reaction network can be mathematically represented by writing mole-balance equations around its metabolites. This leads to a system of differential equations that can be summarized as in the following equation, where $C$ is a vector of metabolite concentrations:

$$\frac{dC}{dt} = f(C) \tag{1}$$

For a system around steady-state, a linear approximation can be made to express the equations in terms of a Jacobian matrix, $J$ (*Steuer et al., 2003*; *Jamshidi & Palsson, 2008*):

$$\frac{dX}{dt} \approx JX \tag{2}$$

with $X = C - C_s$, and $C$ shows concentrations fluctuating around steady-state values, $C_s$. Equation (2) can further be expressed as a Langevin-type equation to explicitly account for small fluctuations (*Steuer et al., 2003*):

$$\frac{dX_i}{dt} = \sum_j J_{ij} X_j + \sqrt{2D_i} \delta_i(t) \tag{3}$$

$$J_{ij} = \frac{\partial \left( \frac{dC_i}{dt} \right)}{\partial C_j} \tag{4}$$

where $D_i$ shows the extent of fluctuation and $\delta_i$ is a random number from unit normal distribution. As demonstrated in the literature (*Kampen, 1992*), Eq. (3) can be written as follows at steady-state:

$$JC + CJ^T = -2D \tag{5}$$

Equation (5) is known as the Lyapunov equation, and it provides a link between the Jacobian matrix of the network and the covariance matrix of the replicate metabolome data (*Steuer et al., 2003*; *Öksüz, Sadıkoğlu & Çakır, 2013*).

The fluctuation matrix (D) accounts for the inherent randomness in the system. The diagonal elements of D reflect the magnitude of fluctuations observed on each metabolite, and the nondiagonal elements can be assumed as zero (*Steuer et al., 2003*). The equation is determined in terms of calculating the covariance matrix (C) while the Jacobian matrix (J) is provided, however, it is underdetermined in the case of calculating J while C is available. This is because there are $n(n+1)/2$ independent entries in C for an n-metabolite system due to the symmetric nature of the covariance matrix, whereas the Jacobian matrix has $n^2$ independent entries. This equation can be rearranged to a standard linear system of equations (*Öksüz, Sadıkoğlu & Çakır, 2013*) and be represented as follows:

$$Aj + 2d = 0 \tag{6}$$

In this equation, A is a square matrix of size $n^2 \times n^2$ derived from the covariance matrix, j is the vectorized form of the Jacobian matrix with size $n^2 \times 1$, and $d$ is the vectorized form of fluctuation matrix with size $n^2 \times 1$, where n is the number of metabolites. *Öksüz, Sadıkoğlu & Çakır (2013)* used Eq. (6) to solve for Jacobian matrix in an optimization platform using Genetic Algorithm. Beside the minimization of the residual of Lyapunov equation, they used sparsity as a rational objective function to select Jacobians from the solution space such that they have a high number of zeros and satisfy Eq. (6) as well. The multi-objective function that simultaneously maximizes the number of zeros (sparsity) in the Jacobian matrix to be determined and minimizes the residual of Eq. (6) can be represented as follows:

$$f = (\text{number of zeros}) \times \lambda - \log 10(\| Aj + 2d \|) \tag{7}$$

The first term in the equation counts for the number of zeros in the Jacobian matrix that needs to be maximized, the second term counts for the residual of Lyapunov equation that needs to be minimized, and in total the objective function $f$ is to be maximized. Lambda ($\lambda$) is a scaling factor discussed in detail in a section below. In order to balance between the two goals in the objective function, and also to refrain the solution from going to a very high number of zeros, the scaling factor was introduced in the objective function.

Using the exact covariance and predefined fluctuation vector as inputs to the algorithm, *Öksüz, Sadıkoğlu & Çakır (2013)* validated the theoretical applicability of this approach. Here, we used the same objective function, but after careful examination of the problem we came up with an extensively modified algorithm that is highly robust and could be applied to the replicates of *in-silico* metabolome data generated by simulating stochastic dynamic models of metabolism using stochastic differential equation (SDE) solvers. The MATLAB function file of the SDE model that was simulated to generate SDE data for yeast is available in Data S1.

Simulations and optimizations were performed in MATLAB (R2017a) on a desktop computer equipped with a 3.2 GHz CPU and 4GB RAM. SDE simulations were performed using the SDE toolbox that is freely available as an external MATLAB toolbox (*Picchini,*

*2007*). Genetic algorithm (GA) was implemented using the *ga* function in MATLAB's global optimization toolbox. A built-in parallelized version of *ga* was used with the help of MATLAB's parallel computing toolbox. Custom MATLAB functions were written for *creation*, *crossover* and *mutation* fields of GA. Maximum number of generations was set to 800 and a mutation rate of 5% was selected after careful examination of GA's behavior. The MATLAB codes for JacLy is available in Data S2.

## Optimization pipeline

Genetic Algorithm (GA) was used to solve Eq. (6) for *j* while A and *d* are settled. At each generation of GA, bit string vectors are generated for *j* as individuals. With a candidate bit string for the structure of the Jacobian vector (zero and nonzero elements in *j*), Eq. (6) can be reduced to a lower dimensional system of equations by removing the zeros in the *j* and also removing the corresponding columns in A.

$$A_r j_r + 2d = 0 \tag{8}$$

Since the Jacobian vector is sparse in structure ($r << n$), this leads to considerable reduction in the number of unknowns to be determined, and increases the speed of the inference algorithm compared to the original algorithm in (*Öksüz, Sadıkoğlu & Çakır, 2013*). In Eq. (8), $j_r$ is the reduced form of Jacobian vector, obtained by removing those elements corresponding to zeros in the suggested individual, and $A_r$ is formed by removing the corresponding columns in A. Equation (8) can be easily treated and solved as a line fitting problem, in which the elements of $j_r$ are factors of the line equation and are estimated to make the best fit to the data. Minimizing the Euclidean norm of this fitting is one of the terms in the optimization objective function defined in Eq. (7). The other objective is to maximize the number of zeros in the Jacobian matrix, considering the fact that metabolic networks are sparse networks (discussed in a section below).

## Constraining the solution space by generating sparse individuals

As the number of metabolites and hence size of the network increases, the solution space expands exponentially and the probability of finding the true candidate for Jacobian vector through a stochastic algorithm decreases significantly. Moreover, the computational time and effort increases dramatically with the size of the network (*Hendrickx et al., 2011*). In these situations, it is very important to constrain the solution space as much as possible if we are going to solve the problem (Eq. (8)) in a manageable time. One way to constrain the solution space meaningfully is to control the generation and reproduction of candidate individuals in GA such that those individuals with unwanted characteristics are not produced to be tested at all. Since metabolic networks are naturally sparse networks, setting a minimum sparsity parameter for the generated individuals can be used as a controlling parameter. This was another novelty in the algorithm compared to the original algorithm (*Öksüz, Sadıkoğlu & Çakır, 2013*).

It is known that metabolic networks are highly sparse, meaning that there are much less interactions (edges) in the network compared to the maximum possible number of edges (fully connected network). We calculated the natural sparsity in several known metabolic

networks and we could see that all tested networks have a sparsity larger than 0.55, so a minimum of 50% was selected as the default value for the minimum sparsity parameter in our algorithm. Just by definition of such a parameter, the solution space to search for non-zero values of Jacobian is greatly reduced. Sparsity parameter was defined as the following:

$$\text{sparsity} \equiv \frac{\text{Total number of possible edges in the network} - \text{number of edges in the network}}{\text{Total number of possible edges in the network}}$$

$$\cong \frac{\text{Number of zeros in the Jacobian}}{\text{Total number of elements in the Jacobian}}$$

Based on this definition, a sparsity value of one will mean a network with not even a single edge whereas a value of zero will correspond to a complete digraph. It must be considered that the sparsity calculated from the Jacobian and the one calculated from the biochemical reaction network are not necessarily the same, since the Jacobian also counts for the regulatory interactions which are absent in the biochemical reaction network, but since the number of regulatory interactions is usually insignificant compared to the number of reactions, the two values are very close.

In order to minimize the computational effort and time, we used the minimum sparsity parameter as the control parameter in the creation of the initial population in GA, and then in the production of individuals at subsequent generations. To this goal, custom MATLAB functions were written and used for creation, crossover and mutation fields of GA in MATLAB. This was another novelty over the previous algorithm (*Öksüz, Sadıkoğlu & Çakır, 2013*). With this supervised control of individuals, bitstrings with unwanted characteristics had no chance to appear as the candidates for Jacobian vector, and it provided a significant contribution in constraining the solution space. The custom functions for GA were written in such a way that minimum sparsity is intrinsic in the generation of individuals and no time is consumed for control and filtering of the generated bitstrings.

## Scanning for the scaling factor

The objective function (Eq. (7)) consists of two terms, one is the residual of Eq. (8) to be minimized and the other is the number of zeros in the Jacobian vector to be maximized. In order to balance between these two values and also to prevent the optimization algorithm from converging to the too sparse solutions, a scaling factor ($\lambda$) is multiplied with the term for the number of zeros in the Jacobian vector. Since this parameter directly affects the magnitude of the objective function, it is important to find a range of lambda that leads to sensible solutions. Selecting very small values for the scaling factor leads to the conditions in which the optimization will not be very sensitive to the number of zeros in the Jacobian vector, and minimization of the residual of Eq. (8) would be dominant in the objective function. On the other hand, large magnitudes of the scaling factor lead to the solutions with very high number of zeros in the Jacobian vector, with almost no sensitivity to the residual of Eq. (8). There is always a narrow interval for the scaling factor, in which the optimization problem can find a Jacobian vector with optimum number of zeros that also leads to insignificant residual value for Eq. (8). This interval for the scaling factor varies

from problem to problem (*Öksüz, Sadıkoğlu & Çakır, 2013*), depending on several factors, among which are the size of the network and the accuracy and number of data replicates from which covariance matrix is calculated. Constant problem-specific λ values were used in the previous algorithm (*Öksüz, Sadıkoğlu & Çakır, 2013*). In order to circumvent the obstacles due to selection of a proper value for the scaling factor, instead of estimating a constant value for each specific problem, we decided to scan a range of values for this parameter in this work in an unsupervised manner. We scanned a range roughly between 0.01–0.10, with increments of 0.005. Since we repeated the algorithm 10 times due to the stochastic nature of genetic algorithm, this led to a total of about 200 solutions per network inference problem. In this way, the optimization algorithm works repeatedly for each value of the scaling factor, and the optimum solution would have chance to appear among the candidate solutions. This was another improvement over the previous algorithm.

### Fluctuation vector

One of the major obstacles in utilizing the Lyapunov equation is introduced by the fluctuation matrix D since it may contain non-observable quantities (*Kügler & Yang, 2014*). The fluctuation matrix plays a critical role in this equation, and the computed Jacobian matrices are highly sensitive to the values in the fluctuation matrix. After a reasonable fluctuation matrix is selected, the problem of solving the underdetermined equation to find the Jacobian can be formulated as an optimization problem.

The existence of a non-zero fluctuation vector is both physically and mathematically meaningful. Fluctuation vector represents the intrinsic noise in the molecular interactions, which are the source of stochasticity in the replicates of data through which we are going to extract information. A non-zero fluctuation vector also prevents the Eq. (8) to have null space, that otherwise would be problematic. On the other hand, it is not very clear how to find and how to set the values for the fluctuation vector in Eq. (8). In a previous work (*Öksüz, Sadıkoğlu & Çakır, 2013*), a constant problem non-specific small value of 0.005 was used for all metabolites to mimic small fluctuations around metabolites. Here, we hypothesize that standard deviation of the data replicates would be an acceptable candidate to be selected as the diagonal elements of the vectorized fluctuation matrix. The use of data-specific fluctuation vector elements in this manner rather than using a constant value for all problems is another improvement in this algorithm compared to the original algorithm (*Öksüz, Sadıkoğlu & Çakır, 2013*).

### Using a community of estimated Jacobians instead of only one elite Jacobian to infer the structure of the network

JacLy scans a range of scaling factors (λ) in the objective function (Eq. (7)). For each scaling factor, optimization is performed 10 times, leading to hundreds of optimizations. The end result of each optimization is a Jacobian vector that has the maximum objective function value among thousands of other individuals. We call this Jacobian as the best-found Jacobian for each optimization. Among all the best-found Jacobians, one can be selected as the elite Jacobian vector based on both sparsity and residual of Eq. (8). In all the test studies, we could observe that if, instead of the elite Jacobian, we combine a community

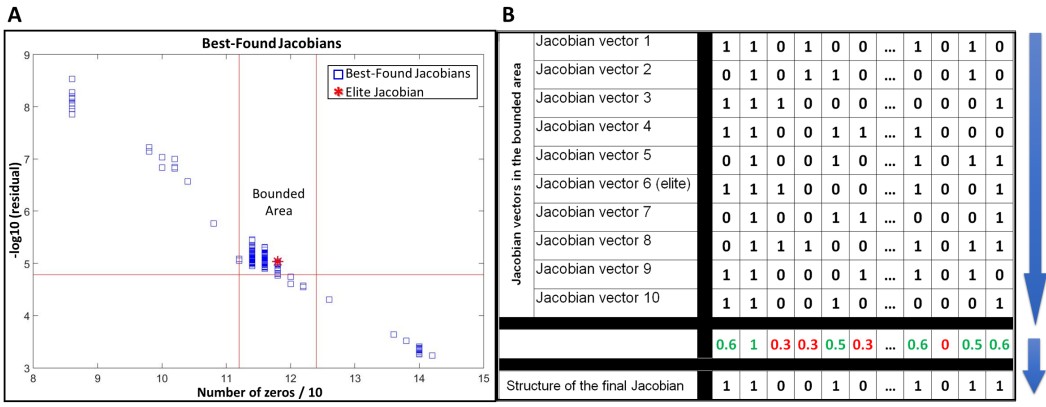

**Figure 1** **Use of a community of the best-found Jacobians to infer the structure of the network.** (A) selecting a bounded area around the elite Jacobian based on a percentage of the number of zeros in the elite Jacobian and the residual of Eq. (8). (B) a schematic example of alignment and combination of Jacobian vectors in the selected community to come up with the final structure.

of the best-found Jacobians in a bounded area around the elite Jacobian and use that community to infer the structure of final Jacobian, the accuracy of inference significantly increases. To do so, first a bounded area is defined around the elite Jacobian based on the number of zeros and the residual of Eq. (8). For all of the tests in this study, ±5% of the number of zeros in the elite Jacobian was selected to set the lower and upper vertical boundaries and −5% of the residual of Eq. (8) for the elite Jacobian was selected to set the lower horizontal boundary (see Fig. 1A). The Jacobian vectors in the bounded area are then binarized by setting their non-zero elements to one. The binarized vectors are then aligned on top of each other to form a binary matrix. Taking average over columns of this matrix leads to a new vector including fractional numbers between 0 and 1. The structure of the final Jacobian vector is then determined by setting a threshold of 0.5 to decide for zero and non-zero values. Those elements that are smaller than the threshold are set to zero and others to one. A looser threshold of 0.4 increases both TPR and FPR. We observed that selecting 0.4 as the threshold leads to better $g$-scores in general, however, there is the risk of sparsity being dropped to significantly lower values. So, we advise the use of sparsity check as a caution while selecting a threshold for the combination of best-found Jacobians. Figure 1B provides a schematic of this procedure.

In this study, we are not providing a mathematical proof of how using a community of best-found Jacobians around the elite Jacobian can improve the inference results. However, as far as we tested with different *in-silico* datasets and noisy covariances, using such combinatorial approach not only leads to better inference results, but also it stabilizes the final output of the algorithm when applied to the same problem repeatedly. The use of a community of best-found Jacobians is another novelty over our previous work (*Öksüz, Sadıkoğlu & Çakır, 2013*), which reported the results based on only the elite Jacobian.

## Quantification of inference performance

While True Positive Rate (TPR) and False Positive Rate (FPR) are two quantities suitable for the evaluation and comparison of network inference results, $g$-score can be used as a single parameter to quantify performance of any inference method. $g$-score is calculated by the following equation:

$$g - score = \sqrt{TPR \times (1 - FPR)}. \tag{9}$$

## RESULTS

### Use of *in silico* Covariance Matrix for metabolic models of *S. cerevisiae* and *E. coli*

The Lyapunov equation (Eq. (5), and Eq. (6) in the rearranged form) is underdetermined in terms of calculating the Jacobian matrix J given C and D as inputs, meaning that there is more than one Jacobian matrix satisfying the Lyapunov equation for each pair of C and D (see 'Methods' section). To evaluate the applicability and performance of our method (JacLy) in predicting the network structure through the prediction of the Jacobian matrix, first we used two kinetic models that are well known in the literature. The first model covers 13 metabolites of yeast glycolysis (*Teusink et al., 2000*), and the second model covers 18 metabolites of central carbon metabolism in *E.coli* (*Chassagnole et al., 2002*). True Jacobian matrix was calculated for each kinetic model around its corresponding steady-state by using the detailed rate expressions and parameters in the models. Here we followed the strategy in our previous work (*Öksüz, Sadıkoğlu & Çakır, 2013*), where we tested our genetic-algorithm-based dual objective formulation in terms of the predictability of the Jacobian matrix by using exact covariance matrices as input rather than deriving covariance matrices, from *in silico* data for example. Our goal in this section is solely to demonstrate the improvements in the current version of our algorithm compared to the previously published algorithm (*Öksüz, Sadıkoğlu & Çakır, 2013*). Having the true Jacobian matrix and predefined fluctuation matrices, the exact covariance matrix was calculated from Eq. (5). We call these covariance matrices as "exact" covariances since they hold true to the Lyapunov equation. Exact covariances and corresponding fluctuation matrices were then used as inputs to JacLy to evaluate its performance in finding J.

JacLy uses genetic algorithm for optimization, which is a stochastic optimization algorithm. Therefore, it is important to solve for the equation for enough number of times until a constant reproducibility parameter is achieved. For both models, a constant reproducibility is obtained after 20 runs. Out of 20 repetitive runs for each model, 19 and 18 of them could find Jacobian matrices that are in complete agreement (100% TPR and 0% FPR) with the true networks of yeast and *E.coli* models, respectively. These results show a great improvement over the previous work (*Öksüz, Sadıkoğlu & Çakır, 2013*), which had a reproducibility parameter of 50% and 5% for yeast and *E.coli* models, respectively. On our desktop computer, each run takes around two minutes for the yeast model and six minutes for *E.coli*, showing a 10 fold increase in computational speed over the previous work (*Öksüz, Sadıkoğlu & Çakır, 2013*). Such significant improvements

in reproducibility and computational speed have been achieved solely by modifying the algorithm and corresponding functions (see 'Methods' section). One should note that since JacLy incorporates a λ scan with 10 replicate solutions, the whole process of generating 200 solutions for *S. cerevisiae* took one hour while the time in the case of *E.coli* was two hours.

We also evaluated the performance of JacLy on noisy covariances. To this goal, we used the exact covariance of the yeast model and followed the same procedure as the previous work (*Öksüz, Sadıkoğlu & Çakır, 2013*) to make noisy data. Random numbers were sampled from a normal distribution with a mean of 1 and a standard deviation of 0.005. This corresponds to a dataset with 50% noise (*De la Fuente et al., 2004*). The random numbers were then symmetrically multiplied with the elements of the exact covariance to generate a noisy covariance matrix. This was repeated to generate ten different noisy covariances and JacLy was applied on each. The average TPR and FPR are 74% and 5%, respectively. These numbers were 73% TPR and 11% FPR in the previous work (*Öksüz, Sadıkoğlu & Çakır, 2013*). These results show a considerable increase in the performance of JacLy compared to its ancestor in terms of the FPR value since exactly the same problem was solved with only improvements in the algorithm based on (i) the use of reduced form of the Lyapunov equation, (ii) the use of sparsity constraint, (iii) scanning the scaling factor and (iv) the use of a community of candidate Jacobian vectors, as discussed in detail in the 'Methods' section. Additionally, note that a threshold of 0.4 in the combination of Jacobian vectors rather than 0.5 leads to a TPR of 84% and an FPR of 8%.

It was reported in the literature that statistical methods such as LASSO and Tikhonov regularization fail to solve Eq. (6) whenever the condition number of matrix A is significantly large (*Sun, Länger & Weckwerth, 2015*). In order to evaluate the sensitivity of our method to the condition number of A, we used different fluctuation matrices along with the true Jacobians of yeast and *E.coli* models as inputs to Eq. (5), and different exact covariances were calculated leading to different A matrices covering a range of condition numbers from $10^6$ to $10^{25}$. JacLy was applied to each of those covariance matrices along with their corresponding fluctuation matrices. We could see that the condition number of A doesn't have any influence on the performance of our algorithm. Even for the largest condition numbers, JacLy was able to find the true Jacobian with similar computational time and reproducibility parameters. It should be kept in mind that not being sensitive to the condition number of A in solving Eq. (6) doesn't mean that the calculated Jacobian matrix is not sensitive to the changes in the fluctuation matrix. Indeed, Eq. (6) is frequently ill-conditioned as it is also reported in other studies (*Sun, Länger & Weckwerth, 2015*). Small changes in the fluctuation matrix D lead to big changes in the calculated Jacobian matrix.

## Use of *in silico* metabolome data for metabolic models of *S. cerevisiae* and *E. coli*

At this stage we applied JacLy to the replicates of *in silico* metabolome data. Stochastic versions of yeast (*Teusink et al., 2000*) and *E.coli* (*Chassagnole et al., 2002*) models were used to generate 1,000 replicates of steady-state metabolome data *in silico*. In this case we had to come up with a fluctuation matrix to be used as input to the method along with

**Table 1** Inference results for the *in-silico* metabolome data, comparison of JacLy and GGM.

| | *In-silico* data for Yeast | | | *In-silico* data for *E.coli* | | |
|---|---|---|---|---|---|---|
| | TPR | FPR | $g$-score | TPR | FPR | $g$-score |
| JacLy | 0.66 | 0.08 | 0.78 | 0.69 | 0.29 | 0.70 |
| GGM | 0.76 | 0.12 | 0.82 | 0.63 | 0.12 | 0.74 |

the covariance of metabolome data. As it was mentioned in the 'Methods' section, we hypothesized that standard deviation of data might be a reasonable source to be used for the construction of a fluctuation matrix. In a stochastic dynamic system all or some of the sources of stochasticity are usually unknown. In the Lyapunov equation the fluctuation matrix D is the parameter counting for sources of stochasticity. Since standard deviation is a measure of variation in data, we used it as a reasonable source to construct the fluctuation matrix. Table 1 shows the inference results of JacLy applied to *in-silico* data for the yeast and *E.coli* with a comparison to GGM-based inference results. In GGM analysis a cut-off of 0.01 was used for *p*-values to decide on the significance of partial Pearson correlation values. The networks predicted by JacLy are directed while those estimated by GGM are undirected.

It must be considered that solving Eq. (6) for the Jacobian vector is highly sensitive to the fluctuation vector d, and so it is of critical importance to come up with a fluctuation vector that is most reasonable for data replicates. We thought of normalization as a way that might improve the correspondence between the covariance matrix C and the fluctuation matrix D in the Lyapunov equation. Data normalization doesn't have any effect on the results of similarity-based inference methods such as GGM. *In-silico* metabolome data for the yeast and *E. coli* were normalized to between 0 and 1 by dividing each value to the maximum value in the dataset. Normalized data was then used to make both covariance matrix C and fluctuation matrix D. When applied to the normalized data, JacLy showed a significant improvement in inference results for the yeast data (0.95 TPR and 0.13 FPR, with a *g*-score of 0.91) while it had no effect on the inference results of *E.coli* data.

Another parameter influential on the result of network inference is the number of replicates in the data. Previous GGM-based analysis for the inference of metabolic interactions using *in silico* metabolome data for the same networks analyzed here showed a sharp decrease in the quality of the inference after the number of replicates decreased below 200 (*Çakır et al., 2009*). Here we tested the effect of number of datapoints on the inference results of JacLy. Of 1,000 replicates initially generated by stochastic differential equations, 100 randomly chosen replicates were used in the inference of the network for *S. cerevisiae*. Repeating this 10 times and taking the average, a TPR of 0.73 and an FPR of 0.16 was obtained by using JacLy, corresponding to a *g*-score of 0.78. On the other hand, GGM-based inference for the same randomly chosen 100 replicates resulted in average TPR and FPR values of 0.42 and 0.03, respectively, with a *g*-score of 0.63. Therefore, an advantage of JacLy over GGM is its considerable robustness in terms of the number of replicate datapoints used in the covariance/correlation calculation.
In the process of inferring a network for a set of metabolites, there are cases when existence (true positive) or non-existence (true negative) of an edge between two metabolites might be available as prior knowledge. Such information can be used as additional input to inference algorithms, resulting in a shrinkage of the solution space and so enhancing the computational speed and performance of the algorithm. We tested the effect of prior knowledge for non-existent edges on the performance of JacLy. To this aim, 7 and 20 zeros of the true Jacobian matrices were selected as priorly known true negatives for yeast and *E. coli* models, respectively. This corresponds to 5% and 7% of the total number of elements in Jacobian matrices for yeast and *E. coli* models. This procedure was repeated 10 times for each model and JacLy was applied to data each time. The average TPR and FPR over 10 repetitions for the yeast model are 0.75 and 0.08, respectively, leading to a *g*-score of 0.83. For *E. coli*, an average TPR of 0.79 and an average FPR of 0.31 was obtained, leading to a *g*-score of 0.74. These results show a significant improvement compared to the corresponding values in Table 1. Based on the results, JacLy performs considerably better when a very small portion of true negatives is introduced as prior knowledge. Specifying true negatives contributed to a better prediction of true positives. The correlation-based GGM approach, on the other hand, is not suitable for the use of prior knowledge.

In addition to the binary structure of estimated Jacobians, which is the main output in inferring the structure of an active metabolic pathway, we also compared the best-found Jacobians in the selected area around the elite Jacobian—before binarization and combination—with the true mechanistic Jacobians of kinetic models calculated by using detailed rate expressions and parameter values in those models. Since JacLy has a stochastic nature, we repeated the optimization on the same SDE data three times, and then we used the Spearman correlation to make the comparisons. The medians of correlations are around 0.45 and 0.25 for yeast and *E. coli* models, respectively while the medians of *p*-values are less than 0.0001 in all cases (See Fig. S1).

We used the kinetic-model-based true Jacobian matrices together with SDE-data-based covariance matrices in the Lyapunov equation to calculate D and see its similarity with our approximation for D. We observed that the calculated D contained off-diagonal non-zero elements as opposed to the approximated D. Some of the elements had the same magnitude as the diagonal elements. On the other hand, our very simple method of estimating D led to quite acceptable TPR and FPR values in those case studies, and the standard deviation of data is logically related to the source of natural fluctuation in the system. Therefore, our estimation approach can be used because of its simplicity and applicability. However, research should be performed to develop a more accurate method of estimating D. On the other hand, one should note that SDE simulator algorithms and toolboxes, such as the one used in this study, have stability problems in terms of the generated noise when applied to highly nonlinear systems. This could also be another reason behind the inconsistency between the covariance of SDE data and the true Jacobian matrix, which directly affects calculation of the fluctuation matrix from the Lyapunov equation.

## DISCUSSION

JacLy is a network inference algorithm with specific focus on the inference of small-scale metabolic networks from steady-state data. It has significant advantages over its ancestor (*Öksüz, Sadıkoğlu & Çakır, 2013*). Here, we reported algorithmic improvements included in JacLy, which led to significant improvements in the runtime and prediction power. Major improvements were (i) vectorizing all possible computations, significantly improving the runtime, (ii) the use of the reduced form of the Lyapunov equation by removing the columns corresponding to zero Jacobian vector entries, improving the runtime, (iii) the use of sparsity constraint in genetic algorithm to improve the runtime by eliminating the possibility of generating low-sparsity individuals, (iv) scanning the scaling factor rather than fixing it for each specific problem, making the algorithm more flexible and independent from the effect of chosen parameter value and improving the prediction power, and (v) the use of a community of candidate Jacobian vectors rather than using the elite Jacobian vector in the inference, improving the prediction power of the results. Inferring metabolic pathways via prediction of Jacobian matrices is also useful in estimating dynamic and mechanistic characteristics of the system under investigation.

One issue that is worth mentioning at this stage is the applicability of the approach in terms of the size of the network to be inferred. For example, each run for the *E.coli* model consumed about twice more computational time compared to that of yeast, while the *E.coli* model has only five more metabolites compared to the yeast model—an almost 40% increase in the number of network nodes. This dramatic increase in computational time with respect to network size—whenever the calculation of Jacobian matrix is involved in a network inference method—was observed and explained in previous studies (*Hendrickx et al., 2011*), and it is indeed one of the major drawbacks of using such methods to infer larger networks. From this aspect, JacLy is more suitable as a small-scale (<20 metabolites) network inference method. There are several network inference methods in the literature with a specific focus on small-scale networks (*Weber et al., 2013*; *Linde et al., 2015*). Since different cellular functions are biologically attached to smaller metabolic pathways or subnetworks, it still makes sense to be able to infer active subnetworks for a specific cellular condition rather than inferring the whole network. Table 2 summarizes some characteristics of JacLy through a comparison with GGM as one of the most common methods in inference of biological networks.

Currently, steady-state metabolome measurement data that are reported in the literature are limited in terms of the number of replicates. This limitation is not specific to our method; commonly used correlation-based inference methods are also suffering from low number of data replicates and usually lead to significantly high number of false positives. Also, in case of real metabolome data, experimental measurement errors interfere with natural stochasticity of the system leading to lower quality in predicted networks. Moreover, since our method relies on the fluctuation matrix (D) as one of its inputs, this external noise is more troublesome. To test the applicability of our approach to the real metabolome data, we introduced random noise to the SDE data of the yeast model by following the approach presented by *De la Fuente et al. (2004)*. For each metabolite, we sampled the random noise

**Table 2  A summarized comparison of JacLy with GGM.**

|  | JacLy | GGM |
|---|---|---|
| Computational time versus network size | NP-hard problem<br>Computational time increases exponentially by increasing the network size | No sensible change in the computational time from very small to very large networks |
| Accuracy versus number of data replicates | Accuracy is a moderate function of number of data replicates<br>For lower number of data replicates, it outperforms correlation based methods | Accuracy is a very strong function of number of data replicates<br>Reduction in the number of data replicates has a very high negative impact on the quality of inferred network |
| Directionality of inferred network | Directed | Undirected |
| Meaningfulness of inferred edge's weights | Mechanistically meaningful<br>Inferred values for the Jacobian elements are measures of interaction strengths and their sign (positive/negative) points into the nature of interaction | Correlation values cannot be used as any physical or mechanistic parameter of the system |

from a normal distribution with mean zero and a standard deviation equal to 10% of its variance in the data. We generated 10 sets of noisy data using this approach. The *in silico* data already includes randomness due to natural stochasticity since it was generated using an SDE simulation toolbox. This random noise was still added to the data to count for the other sources of error and randomness in the data, such as measurement errors. We then applied both Jacly and GGM on the noisy data sets and compared the inference results with that of noise-free SDE data. The average $g$-score dropped from 0.78 (noise-free data) to 0.71 (noisy datasets) for JacLy and from 0.82 to 0.79 for GGM. These results provide a theoretical base for applicability of our approach to real metabolome data that includes other sources of randomness in addition to the natural stochasticity of the system. Although the results for GGM are better than those of JacLy, one should remember that JacLy clearly outperforms GGM for lower number of data replicates (see 'Results' section).

## CONCLUSIONS

Thanks to the improvements introduced to the network inference algorithm, results reported in the previous work (*Öksüz, Sadıkoğlu & Çakır, 2013*) could be obtained much faster, with much higher reproducibility, and with a higher prediction power. In addition, by applying the approach to *in silico* metabolome data, we showed that the use of standard deviation of replicates is a suitable approximation for the fluctuation matrix used as input to the algorithm. However, there might be more sophisticated ways of estimating a fluctuation matrix that better represents the nature of stochasticity in cellular metabolism. Finding more relevant fluctuation matrices for different biological networks can be an altogether separate research topic and can lead to an increase in accuracy and applicability of Lyapunov based inference methods such as JacLy. Also, the power of JacLy was especially obvious when a considerably lower number of replicates were used, or when a small portion of non-existent edges were introduced as prior knowledge. Prediction of the Jacobian matrix from steady-state data is another power of JacLy over GGM since Jacobian matrix is much more informative and biologically relevant in terms of the network structure. Here, we

have introduced JacLy as an algorithm to infer molecular interactions of small networks since the size of matrix A is proportional to the square of the network size, leading to a dramatic increase in the computational time with respect to the network size. Therefore, it should be applied with caution for metabolic systems having higher than 20 metabolites. Additionally, albeit its remarkably better performance for lower number or replicates compared to a correlation-based inference as shown in this work, the use of JacLy for datasets with lower than 100 replicates should be cautioned.

### Funding

This work was supported by the Turkish Academy of Sciences- Distinguished Young Scientists Award Program (TÜBA-GEBIP) and by TUBITAK, The Scientific and Technological Research Council of Turkey (Project Code: 215M201). The funders had no role in study design, data collection and analysis, decision to publish, or preparation of the manuscript.

### Grant Disclosures

The following grant information was disclosed by the authors:
Turkish Academy of Sciences- Distinguished Young Scientists Award Program (TÜBA-GEBIP).
TUBITAK.
The Scientific and Technological Research Council of Turkey: 215M201.

### Competing Interests

The authors declare there are no competing interests.

### Author Contributions

- Mohammad Jafar Khatibipour conceived and designed the experiments, performed the experiments, analyzed the data, prepared figures and/or tables, authored or reviewed drafts of the paper, approved the final draft, writing MATLAB codes.
- Furkan Kurtoğlu performed the experiments, analyzed the data, approved the final draft.
- Tunahan Çakır conceived and designed the experiments, analyzed the data, authored or reviewed drafts of the paper, approved the final draft.

### Data Availability

Corresponding MATLAB files and data are available as Supplemental Files.

### Supplemental Information

Supplemental information for this article can be found online at http://dx.doi.org/10.7717/peerj.6034#supplemental-information.

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
