# Peer review of "JacLy: a Jacobian-based method for the inference of metabolic interactions from the covariance of steady-state metabolome data"

_PeerJ, doi:10.7717/peerj.6034_

## Round 0.1 · original submission · Major Revisions

Both reviewers bring up a number of questions/issues. Please address these as part of the revision.

Reviewer 1 ·

Basic reporting

The ability to infer the Jacobian matrix for a system from metabolomics data is indeed a powerful approach, but the authors have failed to provide sufficient background on how the Jacobian matrix (J) for a system can be derived mathematically from a mechanistic description of the biological system. The mathematical derivation of J from the stoichiometric matrix (S) of the network and the gradient matrix (G) is given by $J = S \cdot G$ where G is a function of how the kinetics of the reaction are described (see DOI: 10.1038/msb.2008.8 or other articles by the same authors for a more complete description). The authors should provide background on this mechanistic approach for deriving J as a complementary approach to the data driven one discussed in this manuscript. Further, the authors could then discuss the obvious benefits of being able to infer J from data instead of deriving it mathematically from a mechanistic model.

Experimental design

(1) With such a plethora of metabolomics data available, the authors should use real data (i.e., experimental values instead of in silico data) to validate their model rather than using an unvalidated model (i.e., in silico data) to validate a model. I understand the need to use sufficiently large amounts of data (i.e., 1000 in silico data sets) to validate a data driven approach such as the one presented here, but the in silico data should complement validation with real data, not replace it.

The authors of the the E. coli model being used (DOI: 10.1002/bit.10288) provide measurements (Table VIII) that could be used to validate the model presented in this manuscript. The authors should then compare the J matrices obtained from network inference and from the measured data.

(2) To the point above regarding the mathematical derivation of J from the structure of a network and metabolomics measurements, the authors should compare their inferred J matrix to the J matrix obtained mathematically. In this context, it is important to recognize that J is not unique for a given system; rather, J depends on the mechanistic detail (i.e., rate laws) used to describe a system (e.g., mass action, Michaelis-Menten; see DOI: 10.1186/s12918-016-0283-2 or DOI: 10.1002/bit.20558 for a discussion). The simplest validation here would be to compare against a model that is based on mass action kinetics (e.g., DOI: 10.1371/journal.pone.0189880), although a more sophisticated model would provide the most accurate representation of network dynamics.

The authors of the E. coli model being used also provide the kinetic rate equations for their model (Table IV); the authors should therefore use this model to derive J and compare it with the J matrix obtained from their network inference method for validation.

(3) The models being used for E. coli and S. cerevisiae here are quite small (18 and 13 metabolites, respectively). Genome-scale metabolic models for these organisms exist with 1,192 metabolites (DOI: 10.1038/nbt.3956) and 2,220 metabolites (DOI: 10.1371/journal.pcbi.1004530), and recent advances in constraint-based modeling allow for network-level dynamics to be studying at the genome-scale (DOI: 10.1038/srep46249) which at least partially alleviate the need for kinetic models. There is still obviously a need for kinetic models of metabolism, but even these models have advanced well beyond the scale of the models used here that are 15+ years old (e.g., E. coli more with 93 metabolites from DOI: 10.1016/j.ymben.2014.05.014) and have even looked at the dynamic characteristics of the network using the Jacobian on the cell-scale (DOI: 10.1371/journal.pone.0189880). Solving these network inference problems are computationally intensive and NP-hard (as the authors note), but is it entirely intractable to apply this method to a model with 93 metabolites (DOI: 10.1016/j.ymben.2014.05.014)? It is unclear from the authors’ description approximately how long simulating a larger model would take (i.e., how does this method scale with respect to time?). In the first Results section, the authors note that it takes 6 minutes to run their software for E. coli, totaling 2 hours for a total of 20 runs; it would be good to provide sensitivity analysis to show the cutoff for when the network becomes so large such that using this method becomes intractable. Methods are available for reducing the size of genome-scale metabolic models (e.g., DOI: 10.1186/s12918-015-0191-x); the authors could use a method such as this to test their method on realistic models of varying size.

Validity of the findings

Overall, the validity of the findings are impossible to assess because of the points raised above regarding the experimental design.

Additional comments

n/a

·

Basic reporting

no comments

Experimental design

no comment

Validity of the findings

no comment

Additional comments

General
An improved method is presented determining the Jacobian matrix (network) from in silico steady state data. The method is significantly faster and better than the previous one. As is mentioned in the paper but not in the abstract the method is only capable of inferring small networks. As input the method (solving the Lyaponov equation) needs a so-called fluctuation matrix. This matrix is approximated with a diagonal matrix containing the standard deviation of the individual metabolites obtained from replicate simulations. The methods and results are in general well described and seem scientifically sound.
The main aim of an inference method would be to use it on measured data. It is not clear to me how to get a fluctuation matrix in that case because the fluctuations would be dominated by measurement errors and not by the randomness of the system itself. I would like the authors to comment on how they expect their method to contribute to the knowledge on metabolic networks.
Detailed comments:
Line 60: Hidden in the data rather than hidden behind the data
Line 61: of that information
139 the – sign on equation 3 has nothing to do with taking the log10 also without taking log10 of the error there has to be a minus sign because the error is to be minimized while the object function is to be maximized. Please rephrase. In my opinion not taking the log10 of the error would result in larger lambda but the same results.
Line 150 I would like more information how the SDE look. This is necessary to reproduce the results.
Line 228 I think there is no correct value for lambda. There is a range of lambda that gives sensible solutions.
Section starting at Line 250
I think for the simulations you can calculate the exact Jacobian as well as the covariance matrix with equation 1 you could calculate the true D of this system I would be interested how it looks like and how this compares to the approximation used here.

The matlab file for the analysis are provided. I would like some more comments without these the script is hard to understand. Only results from the SDE are provided not the files to do the SDE.

---

## Round 0.2 · Major Revisions

While improvements have been made, the reviewers still have significant issues with regard to providing sufficient information such that this approach could be used by others. Along with network size and related pieces of information that should be defined, please also take time to consider the reviewer comments regarding readability.

Reviewer 1 ·

Basic reporting

The authors have addressed several of the major points and clarified the other aspects of the original manuscript which were unclear.

Although the authors have updated the text with additional background and references, the reference list at the end of the article does not reflect these changes; please update the references at the end of the study to reflect the in-text citations.

The manuscript should be edited for English and for readability (e.g., “SDE” is never defined).

Experimental design

The authors' point regarding the current technological limitations and availability of metabolomics data is well taken, however the discussion regarding this point in the text (final paragraph of Discussion) is still lacking. Ultimately, the authors have not provided sufficient information regarding the specifications of Jacly for its use by those studying these metabolic systems; if someone were interested in applying Jacly to study their system, the exact limitations on the size of the network and the required amount of data are unclear. Although the data may not currently exist, the authors should at minimum provide clear expectations for (1) the maximum size of the metabolic network and (2) the minimum number of biological replicates needed in order to apply Jacly to real data if/when it becomes available. Ideally, sensitivity analysis would show how Jacly performance scales with (1) network size and (2) number of replicates, as these pieces of information would be necessary for someone attempting to apply the results from this study to other research.

Validity of the findings

no comment

Additional comments

I would second the comment made by the other reviewer regarding the statement that the abstract should explicitly state that the method is only capable of inferring small networks; perhaps the authors could quantify "small-scale" by providing the size of the network (to my point above):

"...method to infer the metabolic interactions of small networks (<20 metabolites)..."

---

## Round 0.3 · accepted · Accept

Thank you for addressing the reviewer concerns.

#